# A Lieb-like lattice in a covalent-organic framework and its Stoner ferromagnetism

Wei Jiang [1,2], Huaqing Huang[1] & Feng Liu[1,3]

Lieb lattice has been extensively studied to realize ferromagnetism due to its exotic flat band. However, its material realization has remained elusive; so far only artificial Lieb lattices have been made experimentally. Here, based on first-principles and tight-binding calculations, we discover that a recently synthesized two-dimensional $sp^2$ carbon-conjugated covalent-organic framework ($sp^2$c-COF) represents a material realization of a Lieb-like lattice. The observed ferromagnetism upon doping arises from a Dirac (valence) band in a non-ideal Lieb lattice with strong electronic inhomogeneity (EI) rather than the topological flat band in an ideal Lieb lattice. The EI, as characterized with a large on-site energy difference and a strong dimerization interaction between the corner and edge-center ligands, quenches the kinetic energy of the usual dispersive Dirac band, subjecting to an instability against spin polarization. We predict an even higher spin density for monolayer $sp^2$c-COF to accommodate a higher doping concentration with reduced interlayer interaction.

[1] Department of Materials Science and Engineering, University of Utah, Salt Lake City, UT 84112, USA. [2] Department of Electrical and Computer Engineering, University of Minnesota, Minneapolis, MN 55455, USA. [3] Collaborative Innovation Center of Quantum Matter, Beijing 100084, China. Correspondence and requests for materials should be addressed to F.L. (email: fliu@eng.utah.edu)

Lieb lattice, i.e., a two-dimensional (2D) edge-centered square lattice (Fig. 1a) is one of a few lattices hosting exotic flat bands subject to strong correlation effects. It has been proposed to spawn various intriguing quantum states, e.g., ferromagnetism[1,2], superconductivity[3], and topological states[4,5]. More recently, it has been further predicted to hold intriguing optical, electronic, and magnetic states[6–8]. Using photonic, cold-atom, and surface patterning techniques[9–11], several artificial Lieb lattices have been made experimentally. However, a material realization of Lieb lattice has remained elusive, making some of the aforementioned properties inaccessible, such as Lieb-lattice ferromagnetism.

On the other hand, achieving ferromagnetism in organic materials has been an active subject for both fundamental interests in organic magnetism and practical applications as permanent magnets supplementary to their inorganic counterparts[12–14]. More recently, organic magnetic materials have attracted increasing attention for their high spin-wave stiffness, weak spin–orbit coupling, and weak hyperfine coupling, which are appealing properties for spintronic device applications[15–18]. Generally, organic magnetic materials involve transition metal (TM) elements, which give rise to magnetic features due to their partially occupied and highly localized $d$ or $f$ orbitals. However, most organic materials contain only light elements without TM, such as covalent-organic frameworks (COFs)[19,20].

COFs have been extensively studied for their well-known promising applications in gas storage and separation, catalysis, and sensing[21,22], as well as newly predicted exotic electronic properties like Dirac and topological states[23–27]. Not surprisingly, efforts have also been made to induce magnetism in COFs; nevertheless, this has been a formidable challenge, because of their tendency to pair all valence electrons through strong covalent bonds to prevent the formation of unpaired spin-polarized electrons. Therefore, it is very surprising and exciting to see the most recent experimental observation of ferromagnetism in a two-dimensional (2D) $sp^2$ carbon-conjugated COF ($sp^2$c-COF) with high spin density[28]. However, its physical origin is completely unknown. This has motivated the present theoretical study, which surprisingly leads to our discovery of the first material system (the $sp^2$c-COF) realizing a Lieb-like lattice and its Stoner ferromagnetism. This is clearly different from those exotic mechanisms[1,29,30] proposed before for materials comprised exclusively of light elements, such as the flat bands in ideal 2D Lieb and Kagome lattices[1,2,31]. It also differs from systems with default-localized states with a strong Columbic repulsion (or a small hopping $t$), such as the localized orbitals[32,33] in organic compounds[34–36] and nano-carbon structures[37–40]. Apparently, it is also different from magnetism induced by external electrical or magnetic fields[41–43].

Based on density functional theory (DFT) calculation and tight-binding (TB) modeling, we discover that the $sp^2$c-COF represents a kind of Lieb lattice (Lieb-3) with "ideally" one flat band sitting between Dirac bands, arising from the edge-center and corner states, respectively. Distinguished from an ideal Lieb lattice, however, $sp^2$c-COF is coupled with a high degree of electronic inhomogeneity (EI), characterized by a large variation of on-site energy on different lattice sites. By mapping the DFT calculated band structure onto the Lieb-3 TB model, the EI is quantified with a large on-site energy difference and a strong dimerization interaction between the corner and edge-center sites. Consequently, the original dispersive Dirac (top valence) band in an ideal Lieb lattice becomes highly localized, subjecting to an instability against spin polarization upon hole doping. Consistent with experiments, we show that the nonmagnetic insulating $sp^2$c-COF becomes a ferromagnetic (FM) metal upon iodine doping, and the generated magnetization is confined on the corner pyrene ligands. Furthermore, we predict that an even higher spin density can be realized in monolayer $sp^2$c-COF, as the interlayer interaction is eliminated to make the valence band more localized.

## Results

**Lieb lattice with electronic inhomogeneity**. We start from the fundamental TB model analysis of the Lieb lattice. There are four key parameters determining the band structure of a Lieb lattice, i.e., the on-site energy difference $\Delta E$, the dimerization interaction $\delta$ between the corner and the edge-center sites, the nearest-neighbor (NN) hopping $t$, and the next-NN (NNN) hopping $t'$, as indicated in Fig. 1a. For an ideal Lieb lattice with $\Delta E$, $\delta$, and $t'$ all equal to zero, the band structure is characterized with Dirac cones formed at the $M$ point, intersected by a flat band, as shown in Fig. 1b. The NNN interaction $t'$ is known to affect the band dispersion along $\Gamma$-$X$/$Y$ direction[6], which is usually rather small and negligible. On the other hand, the $\Delta E$ and $\delta$ are known to suppress the dispersion of the Dirac bands significantly and lift the band degeneracy at the $M$ point (Fig. 1c, d)[6]. It is important to note that nonvanishing $\Delta E$ and $\delta$ cause a finite variation of local electronic potentials at different lattice sites, namely a finite degree of EI. The larger the $\Delta E$ and/or $\delta$, the larger the EI, which can drastically alter the electronic, and especially the magnetic properties of the Lieb lattice, as we elaborate below.

Specifically, a positive/negative $\Delta E$ isolates the upper/lower Dirac band (see e.g., Fig. 1c for positive $\Delta E$) from the other bands, and a non-zero $\delta$ term fully separates the three bands (see Fig. 1d). Although the band dispersion and the degeneracy at certain $k$ points change with these two parameters, main features of the Lieb lattice remain the same[6]. The Dirac band and the flat band are primarily contributed by the corner and edge-center states, respectively[10], as displayed by the projected density of states (PDOS) in the inset of Fig. 1b. We note that there can be different number ($N$) of sites on edges, and all lead to similar band structures, i.e., Dirac bands intersected by flat bands (see Supplementary Fig. 1). These lattices are named as Lieb-$(2N + 1)$ lattice, where $2N + 1$ is the number of sites per unit cell. Apparently, Fig. 1a shows a Lieb-3 lattice with $N = 1$.

**Material realization of the Lieb lattice: $sp^2$c-COF**. We first analyze the molecular structure of the experimentally synthesized 2D $sp^2$c-COF (Fig. 2a), having a feature of full π-conjugation along both $x$ and $y$ directions[28]. The corner and edge-center sites of $sp^2$c-COF are occupied by pyrene (Py) and 1,4-bis(cyanostyryl)benzene (BCSB) ligands, as highlighted by blue and red ellipses, respectively. Therefore, from the structural point of view, one may consider the $sp^2$c-COF, Py(BCSB)$_2$, to be described by a slightly distorted Lieb-3 lattice. To confirm this, we then carry out an electronic band matching analysis. In Fig. 2b, we show the DFT calculated electronic band structure of the Py(BCSB)$_2$ in the $k_z = 0$ plane. Clearly, the intrinsic Py(BCSB)$_2$ is a nonmagnetic insulator, and the band gap is found to be around 1.0 eV (see Suplementary Fig. 2), which is smaller than the experimentally measured 1.9 eV[28]. This is reasonable as gap size is known to be underestimated by the standard DFT method, which, on the other hand, can reasonably capture the dispersions and orbital compositions of both valence and conduction bands. Therefore, to save time we will use the standard DFT method, as the gap size will not affect our main conclusions about magnetic properties. We note that there is also a noticeable dispersion along the $k_z$ direction, indicating strong interactions between the layers (see Supplementary Fig. 2). Furthermore, from the orbital-resolved projected band structure in Fig. 2b, one sees that the bands around the Fermi level are mainly contributed by the $p_z$

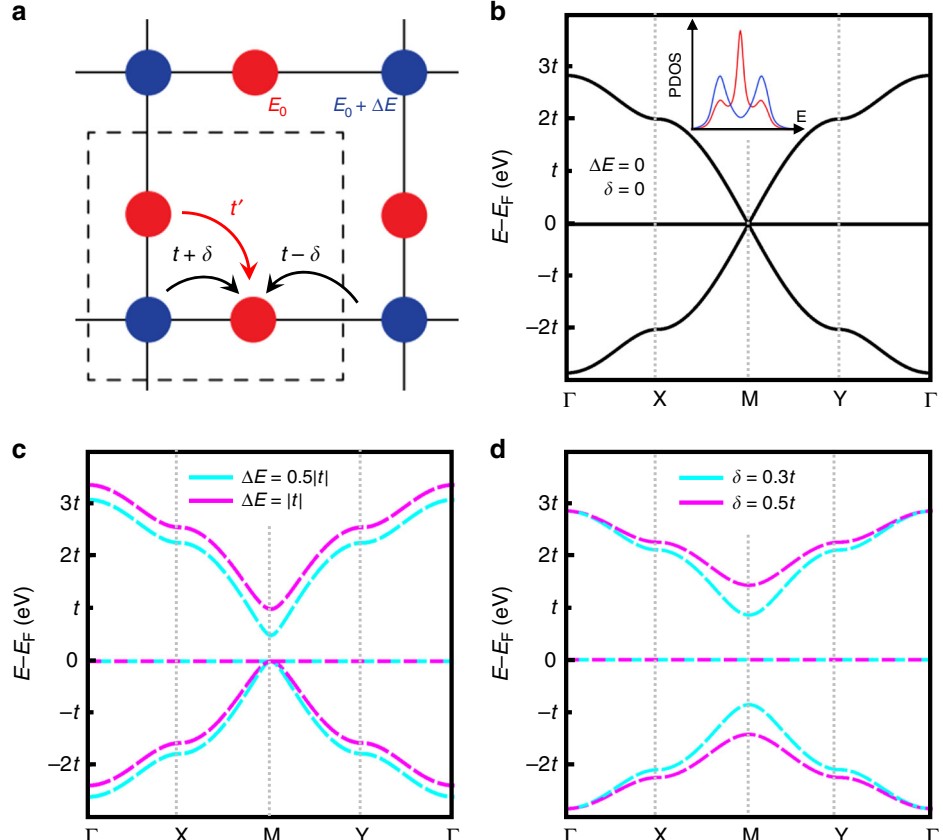

**Fig. 1** Lieb lattice. **a** A Lieb-3 lattice with four key parameters, i.e., the on-site energy difference $\Delta E$, the dimerization interaction $\delta$ between the corner (blue) and the edge-center (red) sites, and the NN and the NNN hopping integral, $t$ and $t'$, respectively. **b** Band structure of an ideal Lieb lattice with $\Delta E$, $\delta$, and $t'$ all equal to zero. The inset shows the projected density of states (PDOS) with blue and red lines corresponding to the corner and edge-center states, respectively. **c** Band structures with different positive $\Delta E$. **d** Band structures with different $\delta$

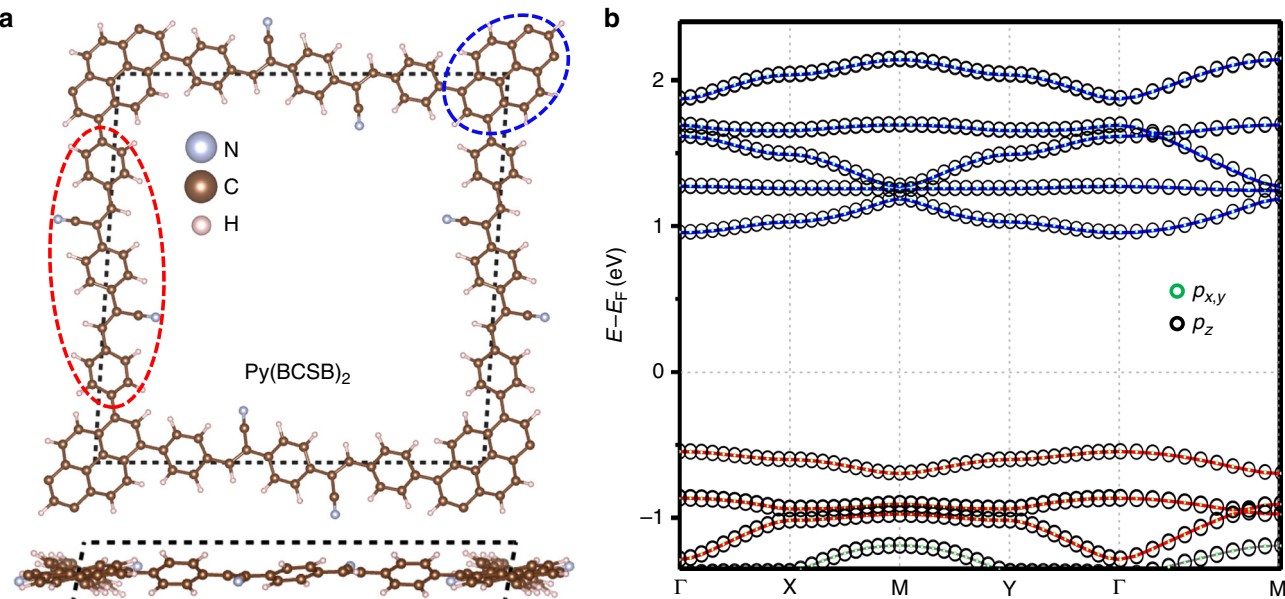

**Fig. 2** 2D COF Py(BCSB)$_2$. **a** Top and side view of crystal structure of Py(BCSB)$_2$ with Py (blue ellipse) and BCSB (red ellipse) ligands sitting on the corner and edge-center sites of the distorted Lieb lattice, respectively. **b** DFT calculated electronic band structure for $k_z = 0$ plane and the orbital-resolved projected band for $p_z$ (black circle) and $p_{x,y}$ (green circle) orbitals. The red and blue bands highlight the Lieb-3 and Lieb-5 band structure, respectively

orbitals, showing the feature of full $\pi$-conjugation formed by the $sp^2$ hybridized C and N atoms. These agree perfectly with the experimental results[28].

We now focus on the top three valence bands (VBs) below the Fermi level, which are actually consistent with Lieb-3 bands, i.e., one nearly flat band in between two Dirac bands, albeit highly perturbed by EI. To reveal this, we calculate the band-resolved charge distribution for these three bands (see Supplementary Fig. 3). We find that the charge distribution for the second band is mostly localized on the BCSB ligands at the edge center, while those for the other two bands are mainly distributed on the Py ligands at the corner. We further performed scanning tunneling spectroscopy simulations by calculating local density of states of edge center and corner sites (see Supplementary Fig. 3), which show a clear feature of Lieb-3 lattice[10]. The large EI exhibited by the Py(BCSB)$_2$ is understandable, because a large $\Delta E$ is expected between the molecular orbitals (MOs) of Py and BCSB ligands and a non-zero $\delta$ is expected from the structural distortion of the Py(BCSB)$_2$ compared with the ideal Lieb-3 lattice. These results strongly suggest that the valence bands are derived from a non-ideal Lieb-3 lattice. In addition, based on the same analysis, we found that the bottom five conduction bands (CBs) above the Fermi level are the Lieb-5 bands (see Supplementary Fig. 4), having two flat bands in between three Dirac bands.

Furthermore, to better understand the mechanism of band formation, it is instructive to study the molecular information of the two building units, i.e., Py and BCSB. As shown in Fig. 3, we found that the MOs around the Fermi level for both Py and BCSB are all $\pi$-conjugated, confirming again the full $\pi$-conjugation of the crystalline Py(BCSB)$_2$[28]. The wavefunctions of the highest occupied MOs (HOMOs) of Py (Fig. 3a) and BCSB (Fig. 3b) show

exactly the same shape as the band-resolved charge distribution for the Lieb-3 VBs (see Supplementary Fig. 3), indicating the electronic Lieb-3 VBs are constructed by the HOMOs of the corner Py and edge-center BCSB ligands. Similarly, the one lowest unoccupied MO (LUMO) of Py and the two LUMOs of BCSB with two localized charge centers each (highlighted by red ellipses in Fig. 3b) form the Lieb-5 band (see Supplementary Fig. 4). This shows a rare coexistence of both Lieb-3 and Lieb-5 bands in one single lattice, which is caused by a coincidence of the energetically perfectly aligned MOs between Py and BCSB ligands. We have also calculated molecular information using Gaussian package showing consistent results (see Supplementary Fig. 5). To more concretely confirm the Lieb-lattice-like nature of the Py(BCSB)$_2$, we performed the maximally localized Wannier functions fitting using the Wannier90 package[44]. The fitted band structure and the corresponding maximally localized Wannier functions show good consistency with the above DFT calculation results and TB analyses below (see Supplementary Fig. 6).

Next, we quantify the degree of EI in Py(BCSB)$_2$ by fitting the TB model to the DFT band to estimate $\Delta E$ and $\delta$. Because the three VBs and five CBs of Py(BCSB)$_2$ are formed by different MOs and separated distinctly in the energy space, they can be independently reconstructed by a three- and five-band TB model on the Lieb-3 and Lieb-5 lattice, respectively. We then adjust the $\Delta E$ based on the energy diagram calculated for MO analysis, and fit the DFT calculated bands with different $t$ and $\delta$ (see Supplementary Fig. 2 and Supplementary Note 1). Specifically for the Lieb-3 band, we found the hopping integral ($t \approx 0.13$ eV) is rather small, which is caused by a large separation between the HOMOs of Py and BCSB ligands. More importantly, we found relatively a large on-site energy difference ($\Delta E \approx t \approx 0.14$ eV) and a

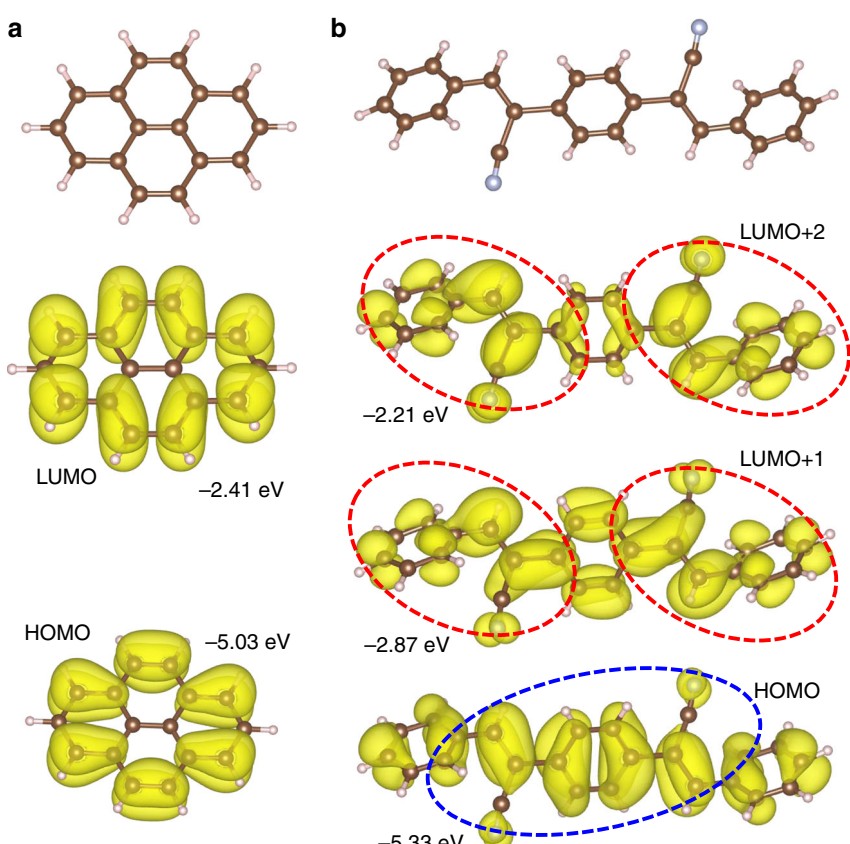

**Fig. 3** Molecular information. MO energy levels and the associated wavefunctions of **a** a Py and **b** a BCSB molecule. The red and blue ellipses highlight the charge localization of LUMOs and HOMO, respectively

strong dimerization interaction ($\delta \approx 0.3t \approx 0.04$ eV) between the two HOMOs, indicating a high degree of EI, giving rise to a very narrow bandwidth ($W \approx 0.36$ eV). Consequently, the upper Dirac band becomes highly localized and is fully isolated from the two bands below. We then compared our TB fitting parameters with those of the Wannier fitted Hamiltonian, which shows a very good agreement, further confirming the strong EI effect to induce a localization of the VB (see Supplementary Note 2).

**Ferromagnetism in $sp^2$c-COF.** Knowing the $sp^2$c-COF to be a Lieb-lattice-like system, we now proceed to explain the physical mechanisms underlying the experimentally observed magnetic behaviors of the $sp^2$c-COF upon iodine doping. It is well known that a Lieb lattice can spawn intriguing magnetic properties associated with the exotic flat band[1,2]. However, the flat band in Py(BCSB)$_2$ is the second VB, which is beyond the reach of typical doping level. Therefore, one must look for a distinct mechanism. We realize that based on Stoner criterion[45], the ferromagnetism can arise from any localized band under partial filling. A closer look at the first VB of Py(BCSB)$_2$ reveals that this usual dispersive Dirac band has become highly localized due to the strong EI. Because the VB is made of $2p$ state, the Stoner parameter is in fact even larger than the $3d$ state[46]. Consequently, one may expect

that this band will now be subjecting to an instability against spin polarization upon hole doping. To confirm this, we carry out a computational experiment to dope the system with one hole per unit cell, and calculate the corresponding electronic and magnetic properties.

From the band structure and PDOS plot shown in Fig. 4a, one can clearly see that the insulating intrinsic Py(BCSB)$_2$ becomes metallic after doping (see Supplementary Fig. 7). The half-filled spin-degenerate bands spontaneously split (spin splitting $J \approx 20$ meV), giving rise to an itinerant FM state. From the difference of charge distribution before and after doping (see Supplementary Fig. 8), the doped holes are found to be mainly located on the corner-site Py ligands with the same shape as HOMO of Py, which is the one to form the Dirac band of the Lieb-3 lattice (inset of Fig. 1b and Supplementary Fig. 3). Consistently, the FM aligned spins are primarily localized on the Py ligands as well, as shown by the spin distribution plot in Fig. 4b. We further calculated the energy of anti-ferromagnetic state to confirm the FM ground state. After extracting the energy difference and fitting to the simplified Heisenberg model (see Supplementary Fig. 9 and Supplementary Note 3), the Curie temperature is estimated to be ~9.3 K, which agrees very well with the experimental results (~8.1 K). Using non-collinear spin calculations considering spin–orbit coupling, we found the magnetic

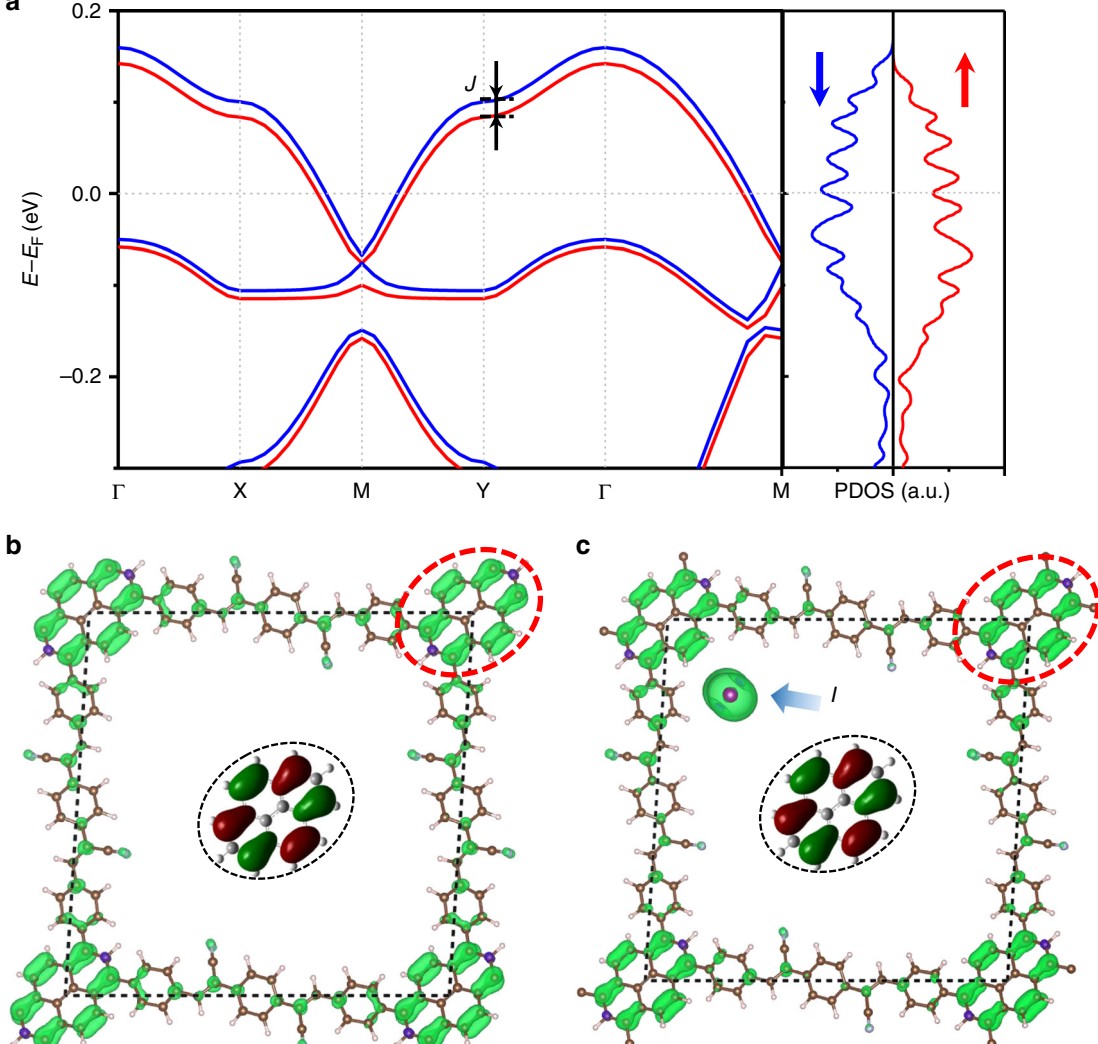

**Fig. 4** Magnetism in Py(BCSB)$_2$. **a** Spin-polarized electronic band structure and PDOS of the hole-doped Py(BCSB)$_2$. Spin distribution of **b** the hole-doped and **c** the iodine-doped Py(BCSB)$_2$ with iodine (highlighted by blue arrow) sitting at the off-corner site. Inset shows the HOMO of Py

moments prefer to align in the direction perpendicular to the plane (see Supplementary Note 4).

In experiments, the $Py(BCSB)_2$ is chemically oxidized (hole-doped) using iodine vapor[28]. Therefore, besides the computational experiment of hole doping, we perform calculations of the $Py(BCSB)_2$ with iodine doping using several representative structural configurations. To decouple the iodine ions between the layers, we construct a two-layer supercell with one iodine. This gives a comparable doping concentration (0.5 hole per unit cell) with the experiments (0–0.7 hole per unit cell)[28]. The off-corner site is found to be the most stable position (Fig. 4c) after examining several possible positions for the iodine (see Supplementary Fig. 10 and Supplementary Note 5), which is used to calculate the electronic and magnetic properties. From the charge difference before and after the iodine doping (see Supplementary Fig. 8), we find a charge transfer from the $Py(BCSB)_2$ to the iodine. The generated holes are mainly distributed on the corner Py ligands. Again, the ground state of the iodine-doped $Py(BCSB)_2$ is found to be FM and the corresponding spin distribution plot is shown in Fig. 4c. We note that besides the magnetization contributed by Py ligands, the iodine also carries certain amount of magnetization due to a partial filling of its $p$ orbitals. We have also examined another two structural configurations with the iodine sitting at the off-edge-center and corner site that has ~0.04 and ~0.36 eV per supercell higher energy, respectively, showing similar magnetic behavior (see Supplementary Fig. 11).

**Magnetization enhancement.** In general, the narrower the bandwidth is, the stronger the magnetization will be[45]. Recall that there is a noticeable dispersion along $k_z$ direction in the band structure of bulk $Py(BCSB)_2$ (see Supplementary Figs. 2 and 7) caused by interlayer interactions, which increases significantly the width of the Dirac band. Interlayer interaction can be eliminated in a monolayer $Py(BCSB)_2$, and hence one may expect enhanced magnetization. To test this idea, we calculate the electronic and magnetic properties of the monolayer $Py(BCSB)_2$, and its three key properties, i.e., bandwidth ($W$), spin splitting ($J$), and the highest magnetization are listed in Supplementary Table 1 in comparison with the bulk. We find that $W$ of the Dirac band is much smaller than that of bulk $Py(BCSB)_2$ and the $J$ is significantly enlarged, which result in a much higher magnetization upon hole doping (see Supplementary Fig. 12).

Another interesting point is that the magnetization of a flat or highly localized band depends critically on band filling. Therefore, it is natural to anticipate the magnetization to change with different doping level. We have calculated the magnetization for both bulk and monolayer $Py(BCSB)_2$ as a function of doping level (see Supplementary Fig. 12). One can clearly see the magnetization increases first at low doping level of holes when only one spin channel becomes partially filled until reaching a maximum, and then decreases at high doping level when holes start to fill both spin channels. Since the experiments only observed an increase of magnetization without any decrease[28], we believe the induced magnetization has not reached the maximum for the iodine-doped $Py(BCSB)_2$. This is possibly because a limited amount of iodine can be doped into the bulk sample with relatively a small surface area. To increase the doping concentration, one may use a thinner or monolayer $Py(BCSB)_2$ with higher surface-to-volume ratios, leading to enhanced magnetization. To test these ideas, we suggest experiments to further study the magnetization of $sp^2$c-COF as a function of doping level and film thickness.

It is worth mentioning that another COF with C=N conjugation (C=N-COF), synthesized by the same group, also shows similar but weaker magnetic response[28]. Our calculations show similar band structure and magnetic behavior upon hole doping for C=N-COF, but interestingly, a stronger magnetic response than the monolayer $Py(BCSB)_2$ (see Supplementary Fig. 13), because of both a narrower $W$ and a larger $J$ (see Supplementary Table 1). This difference could be caused by different sample qualities of these two COFs. We believe a better sample quality may yield an even higher spin density in the C=N-COF. Another interesting point to mention is that if the system could be doped further, the exotic flat band triggered ferromagnetism would evolve, which could be potentially used to study the flat-band-related topological state in Lieb lattice.

In conclusion, we have discovered that the experimentally synthesized 2D organic system of $sp^2$c-COF is a material realization of a Lieb-like lattice. Furthermore, we demonstrate that it is a non-ideal Lieb lattice with strong EI, so that it exhibits an insulator-to-metal transition and a nonconventional magnetic instability, consistent with experimental observation. Our findings open the door to exploiting the highly tunable COFs as a versatile material platform to explore exotic electronic and magnetic properties hosted by Lieb lattices. Moreover, the Stoner mechanism for ferromagnetism provides a useful theoretical guidance in search of new COF-based organic magnets. The advancement in fundamental knowledge enabled by our theoretical study may also be transferred to technological applications for COFs as electronic and spintronic materials, beyond the already-known traditional applications as structural materials. Our work may also foster new research directions to study other exotic physics of Lieb lattices beyond magnetism in COFs, such as topological properties.

## Methods

**DFT calculations.** Our first-principles calculations were carried out within the framework of the Perdew–Burke–Ernzerhof generalized gradient approximation (PBE-GGA)[47], as embedded in the Vienna ab initio simulation package code[48]. All the calculations were performed with a plane-wave cutoff energy of 500 eV. For the $sp^2$c-COF, we adopted the experimental lattice constants and transferred to the primitive cell with $a = b = 24.74$ Å, $c = 3.73$ Å, $\alpha = \beta = 100.09°$, and $\gamma = 91.69°$ [full structural relaxation would only change the lattice constants negligibly (<0.1%)]. For C=N-COF without available experimental results, the structure was fully relaxed, giving rise to lattice constants of $a = b = 24.44$ Å, $\alpha = \gamma = 90.00°$, and $\beta = 99.31°$. To eliminate the interlayer interaction, we introduced a vacuum layer of 20 Å thickness for monolayer calculations. Both $sp^2$c-COF and C=N-COF are reported to have AA stacking experimentally, which will be used for our study of both bulk and bilayer systems. The geometric optimizations were performed without any constraint until the force on each atom is <0.01 eV·Å$^{-1}$ and the change of total energy is smaller than $10^{-4}$ eV per unit cell. The $\Gamma$ centered Brillouin zone $k$-point sampling was set with a spacing of $0.03 \times 2\pi$·Å$^{-1}$, which corresponds to $3 \times 3 \times 11$, $3 \times 3 \times 5$, $2 \times 2 \times 11$, and $3 \times 3 \times 1$ $k$-point meshes for bulk unit cell, $1 \times 1 \times 2$ bilayer supercell, $\sqrt{2} \times \sqrt{2} \times 1$ supercell, and monolayer calculations, respectively. The molecular orbital information of Py and BCSB were calculated using both VASP with PBE-GGA potential and Gaussian package[49] with the B3LYP functional[50]. The computational experiment of hole doping was performed by changing the total number of electrons of the system while maintaining the charge neutrality with a compensating homogenous background charge. Simulations of $Py(BCSB)_2$ with iodine doping were carried out using several representative structural configurations, e.g., corner and edge-center sites. To decouple the iodine ions between the layers, a two-layer supercell with one iodine was applied. Spin–orbit coupling effect of iodine is tested to have negligible effect.

## Data availability

The data that support the findings of this study are available from the corresponding author upon reasonable request.

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

## Acknowledgements

This project is supported by U.S. DOE-BES (Grant No. DE-FG02-04ER46148). W.J. is additionally supported by the National Science Foundation-Material Research Science & Engineering Center (NSF-MRSEC grant No. DMR-1121252). We also thank the CHPC at the University of Utah and DOE-NERSC for providing the computing resources.

## Author contributions

W.J. and F.L. conceived the idea and designed the project. W.J. performed calculations and analysis. W.J. and F.L. wrote the paper. H.H. discussed and commented on the manuscript. All data are reported in the main text and supplementary materials.

## Additional information

**Competing interests:** The authors declare no competing interests.

