## [Peer Review File · Nature Communications]

Reviewers' comments:

Reviewer #1 (Remarks to the Author):

The authors study the origin of magnetism in a crystalline covalent organic framework upon doping. The goal is to provide a theoretical interpretation of recent experimental evidence of ferromagnetism in this system. The idea is to merge ab-initio DFT calculation with tight-binding modelling in terms of Lieb lattice.

Overall, the physical analysis in the paper is not convincing and I do not recommend publication of this manuscript.

The main goal of the paper is to show that "that the nonmagnetic insulating sp²c-COF becomes a ferromagnetic metal upon iodine doping ... and to predict that "an even higher spin density can be realized in monolayer sp²c-COF, as the interlayer interaction is eliminated to make the valence band more localized". Both these results are related to band filling easily obtained in standard ab-initio band structure calculation and, contrary to what the authors say, there is no need – and no advantage – in resorting to models.

I have many misgivings about the two approaches (DFT and TB/Lieb lattice) and about the need to merge them. Here my comments:

1) DFT is a powerful method that can be used in many different "flavors" resulting in different results. Many packages exist that adopt different choices concerning basis set (plane waves, localized orbitals) different functionals etc. These details should be specified. Moreover, was iodine spin-orbit included? Which kind of spin polarized DFT was used?

2) The concept of "local electronic chemical potential varying from site to site" is obscure-meaningless: chemical potential is defined for a system as a whole, not for its parts.

3) Is the "two-layer supercell with one iodine" comparable with the experimental situation? And in general how does theoretical doping compare with the experimental one? The authors mention to have tried different absorption sites: more details on them are necessary, especially when describing high doping.

4) The authors claim to be able to reproduce the calculated band structure with few TB parameters ending up however with a different set of parameters for the valence bands (Lieb-3) and for the conduction bands (Lieb-5). But Lieb-3 and Lieb-5 provide both valence and conduction bands (as shown in fig S1) and it is not possible to use them separately. Any mapping must be unique and reproduce all the bands, not a portion of them.

5) The authors rely on Lieb lattice mapping in order to explain ferromagnetism: Lieb theorem (reference 20, 30,31) refers to Hubbard model where ferromagnetism in 2D is a true many-body effect. How this relates to the one-particle (DFT or TB) approach?

In conclusion, I think that a standard but much more accurate analysis of DFT results will be enough to explain the origin of spin unbalance upon doping. After a deep revision, the paper should be submitted to a more appropriate journal.

To this end I add some minor comments:

a) Attributing the name U to the energy separation between up and down bands is misleading (U is for on-site e-e repulsion, better to use J).

b) The sentence "it is well known..." that appears many times in the manuscript should be accompanied with appropriate references

c) Figure 1a and 1b resembles too much fig 1. Of ref. 37

Reviewer #2 (Remarks to the Author):

The authors presented theoretical investigation of the organic magnets with ferromagnetism in a two-dimensional sp² carbon-conjugated covalent-organic frameworks (sp²c-COF) upon iodine doping without transition metals. By using the first-principles and tight-binding calculation, they

show that the sp²c-COF belongs to a Lieb lattice with strong electronic inhomogeneity (EI), as characterized with a large on-site energy difference and a dimerization interaction between the corner and edge-center ligands. The EI drives the usual dispersive Dirac (valence) band corresponding to the ideal Lieb lattice to become highly localized, and hence subjecting to an instability against spin polarization upon hole doping. Higher spin density for thin film and monolayer sp²c-COF to accommodate a higher doping concentration with reduced interlayer interaction is predicted. I would suggest publication of the manuscript in Nature Communication. Some comments are provided below for the authors consideration.

The organic ferromagnetic COF should have a finite Curie temperature, the authors may discuss its magnitude or compare it with the experimental report in the literature.

Whats the direction of the magnetic moments in the organic magnets?

The found a higher spin density can be realized in monolayer sp²c-COF, as the interlayer interaction is eliminated to make the valence band more localized. They also note that there is also a noticeable dispersion along the k_z direction, indicating strong interactions between the layers, which reduce the band gap to around 1.0 eV. Did the authors consider the stacking effect carefully? I guess a simple AA type stacking is used in the present work, could this simplification be justified? Same question can be asked for both cases of the bilayer and bulk structures. Its interesting to study the enhanced magnetization upon hole doping, one may wonder what the results will be if the system is electron doped.

Page 12, should B3LTP functional be B3LYP functional?

Reviewer #3 (Remarks to the Author):

The manuscript "Organic Ferromagnetism in a Covalent-Organic Framework" describes a purely computational study of magnetism in a covalent organic framework. The fascinating aspect about this material is that it is two-dimensional, built up by sp² carbon-conjugation (sp²c), and contains NO transition metals. The manuscript itself is well-written and the information is clearly and well-presented. The authors find that this material (referred to as sp²c-COF) can be mapped onto a Lieb lattice, which is a theoretical two-dimensional edge-centered lattice for which only artificial realizations exist. Performing density functional theory calculations, the authors can then explain the origin of magnetism in this material upon iodine doping. I do appreciate the layout of the manuscript, where the authors first provide the necessary background about the theoretical Lieb lattice and then show how this relates to the material in question. Only then do they show how its magnetic properties follow upon whole-doping with iodine, making it very easy to follow the arguments of the authors.

I have some comments on the novelty of this work and some technical comments, which I will describe further below. The material itself and its ferromagnetic properties are not novel---they have been described in Science 357, 673 (2017) and I assume this paper has inspired the authors to study this material. However, this Science paper does not provide any insight or even suggestions as to the mechanism that causes this unconventional magnetism, but purely reports its finding. As such, I do see three aspects that are novel: (i) The authors find that sp²c-COF is a realization of a Lieb lattice. All previously found realizations for this type of lattice are somewhat artificial, using photonic, cold-atom, or surface patterning techniques. (ii) The authors can explain the origin of this unconventional magnetism in this materials. And (iii), the authors predict that a monolayer of this material (rather than the experimentally already synthesized bulk material) will

exhibit even stronger ferromagnetism due to reduced inter-layer interactions. All three points are interesting in-and-of-themselves, but the authors also argue that the understanding of such magnetism in organic materials may lead to further developments and practical applications--- although the language is somewhat vague and maybe more of a conjecture. Since this manuscript has several distinct novel aspects, it may appeal to a broader audience.

I also have a few more technical concerns:

* On page 5, the authors refer to the band-structure in Fig. 2b and say that “Clearly, the intrinsic Py(BCSB)₂ is a non-magnetic insulator with a gap around 1.5 eV”. I assume the authors have deduced the gap from a DOS, as that information could not be reliably obtained from the band-structure in Fig. 2b. Also, the magnetic state could not be deduced from the band-structure, unless a spin-resolved band structure would be plotted.

* The authors seem to only have considered collinear spin arrangements. It is conceivable that non-collinear spin arrangements may play a role?

* As the authors mentioned, DFT is not the correct level of theory to deduce the band gap (at least using PBE). But, why was not a post-DFT method applied, i.e. GW, to at least test the validity of the band structure and orbitals in a few cases? GW (G₀W₀) is implemented in VASP and the size of system might allow such calculations. Although, I would assume that the shape of the conduction bands is probably reasonably captured by the DFT calculations the authors performed.

* On page 6, the authors say that “we have qualitatively confirmed that the three valence bands [...] are resulted from the Lieb lattice”. “Confirmed” might be an overstatement here. I would use language such as “The data suggests that the valence bands are the result of a Lieb lattice”. The band structure in Fig. 2b is significantly distorted from the ideal Lieb lattice such that other interpretations are certainly possible.

* From a computational perspective, I am not certain why the authors switched codes for different aspects of the project. The main calculations were performed with VASP, but the calculations on the building blocks of the Lieb lattice, i.e. the Py and BCSB molecules, are performed with Gaussian. These are two very different codes (plane waves vs. Gaussian basis functions) and (PAW potentials vs. all electron). Switching codes introduces unnecessary inconsistencies and different sources or errors that may not cancel anymore. It seems perfectly reasonable that the Py and BCSB molecules would have been modeled in VASP.

* Along the same lines as the previous point---why was the exchange-correlation functional switched between calculations? For the material itself, PBE was used. On the other hand, for the Py and BCSB molecules, B3LYP was used. This introduces inconsistencies and, at best, makes the results uninterpretable.

* On page 10, the authors refer to Fig. S8, but I assume they meant Fig. S7.

* Why was the unit cell for sp²c-COF not relaxed while the one for C=N-COF was?

I assume that the technical points can be overcome by the authors so that the deciding factor will be one of novelty. As pointed out above, the main point of magnetism in sp²c-COF is not novel, but other points made by the authors are.

Response to reviewers' comments

We thank all three reviewers for reviewing our paper. Overall, we found their comments very constructive, which have helped us to further improve the quality and clarity of our paper. Below we respond in detail to their specific comments in a point-by-point manner.

Response to Reviewer #1:

Comment #1: *The authors study the origin of magnetism in a crystalline covalent organic framework upon doping. The goal is to provide a theoretical interpretation of recent experimental evidence of ferromagnetism in this system. The idea is to merge ab-initio DFT calculation with tight-binding modelling in terms of Lieb lattice. Overall, the physical analysis in the paper is not convincing and I do not recommend publication of this manuscript.*

The main goal of the paper is to show that “that the nonmagnetic insulating sp^2c -COF becomes a ferromagnetic metal upon iodine doping ... and to predict that “an even higher spin density can be realized in monolayer sp^2c -COF, as the interlayer interaction is eliminated to make the valence band more localized”. Both these results are related to band filling easily obtained in standard ab-initio band structure calculation and, contrary to what the authors say, there is no need – and no advantage – in resorting to models. I have many misgivings about the two approaches (DFT and TB/Lieb lattice) and about the need to merge them. Here my comments:

Reply: We respectfully disagree with this reviewer on this point, and we are sorry that this reviewer's point of view is possibly caused by our failure to clearly present the novel points of our work as the reviewer #3 has pointed out to us. To clarify, our goal is not simply carrying out DFT calculations to show that “*the nonmagnetic insulating sp^2c -COF becomes a ferromagnetic metal upon iodine doping ... and to predict that “an even higher spin density can be realized in monolayer sp^2c -COF, as the interlayer interaction is eliminated to make the valence band more localized”*”, but instead to reveal **the underlying physical mechanism for the experimental observation of ferromagnetism (FM) in this special sp^2c -COF**. In doing so, we discovered that the sp^2c -COF represents the first material realization of Lieb lattice, which is the most novel point of our work. Furthermore, the observed FM involves an unconventional mechanism different from the flat-band mechanism commonly known for the Lieb lattice. Of course, DFT calculations, as the reviewer has suggested, are able to predict the FM ground state as we show in agreement with experiment, but doing DFT alone cannot tell why FM occurs in this COF. In other words, by doing the DFT calculations alone, one would get no idea this COF having a Lieb lattice and the

physical origin of its FM. It is for this reason that we take an approach of combining the DFT calculations with the TB models to reveal that this special sp^2 c-COF actually has a non-ideal Lieb lattice, and the doping induced FM arises from a high degree of electronic inhomogeneity that makes the otherwise dispersive Dirac band of an ideal Lieb lattice become localized. Apparently, this important finding could not be obtained by doing only the DFT calculations as the reviewer suggested.

Comment #2: *DFT is a powerful method that can be used in many different “flavors” resulting in different results. Many packages exist that adopt different choices concerning basis set (plane waves, localized orbitals) different functionals etc. These details should be specified. Moreover, was iodine spin-orbit included? Which kind of spin polarized DFT was used?*

Reply: We agree that DFT is a powerful tool, but it has also limitations because in a way it can be used like a black box without telling people why one gets the calculation result as it is. We thank the reviewer for the suggestion to add more details of calculation methods, which we have done in the revised manuscript. We have also tested spin-orbit coupling effect of iodine, which has negligible effect to our results. We were using the collinear spin-polarized DFT calculation. In the revised manuscript, the non-collinear spin-polarized DFT calculations have been added and discussed (see also our response to reviewer #2 and #3 on this same point).

Comment #3: *The concept of “local electronic chemical potential varying from site to site” is obscure-meaningless: chemical potential is defined for a system as a whole, not for its parts.*

Reply: We respectfully disagree with the reviewer on this point. To our understanding, besides the “mean” chemical potential, the concept of “local chemical potential” can be well defined, such as in a mass diffusion equation, the chemical potential is generally written as a function of concentration which is in turn a function of “position”. It is the “gradient” of chemical potential that drives the mass flow. In TB formalism as we used here, different on-site energies are commonly used to account for different electronic chemical potential, at different atomic sites.

Comment #4: *Is the “two-layer supercell with one iodine” comparable with the experimental situation? And in general how does theoretical doping compare with the experimental one? The authors mention to have tried different absorption sites: more details on them are necessary, especially when describing high doping.*

Reply: We thank the reviewer for bringing up this point. The two-layer supercell with one iodine we used is comparable with the experimental situation, as the hole concentration in our model (0.5 /unit-cell) is within the experimental doping range (0~0.7 /unit cell). In general, DFT calculations of doping effect has been widely applied in comparison with experiments, such as in the studies of effects of doping in semiconductors. For our study, we tested different adsorption sites and used the three most stable ones with a reasonable doping concentration (0.5h/unit cell) to confirm our model analysis and show the qualitatively agreement with experiments. High doping concentration studies were performed by changing the total number of electrons of the system while maintaining the charge neutrality with a compensating homogenous background charge. We have added more details of the doping calculations, as the reviewer suggested.

Comment #5: *The authors claim to be able to reproduce the calculated band structure with few TB parameters ending up however with a different set of parameters for the valence bands (Lieb-3) and for the conduction bands (Lieb-5). But Lieb-3 and Lieb-5 provide both valence and conduction bands (as shown in fig S1) and it is not possible to use them separately. Any mapping must be unique and reproduce all the bands, not a portion of them.*

Reply: We thank the reviewer for the comment. It seems the reviewer has misunderstood our Lieb-lattice models. The models we show in Fig. S1 for “ideal” Lieb-3 and Lieb-5 lattices contain both valence and conduction bands, by assuming an exact half filling of the respective lattice bands. Instead, for the real material sp^2c -COF, the Lieb-3 bands are fully occupied while the Lieb-5 bands are fully unoccupied, but they are two separate sets of bands coming from different orbitals (*i.e.*, they do not correspond to each other as bonding and antibonding orbitals). This can be clearly seen from our molecular information analysis as shown in Fig. 3 and Fig. S5. The Lieb-3 bands are formed by HOMOs of Py and BCSB, while Lieb-5 bands are formed by LUMOs of Py and BCSB. These two subsets of bands have negligible interactions between them. Therefore, they can be treated and fitted independently.

To avoid possible confusion, we added the following sentence before the TB model: “*Because the three VBs and five CBs of Py(BCSB)₂ are formed by different orbitals and separated distinctly in the energy space, they can be independently reconstructed by a three- and five-band TB model on the Lieb-3 and Lieb-5 lattice, respectively.*”

Comment #6: *The authors rely on Lieb lattice mapping in order to explain ferromagnetism: Lieb theorem (reference 20, 30,31) refers to Hubbard model where ferromagnetism in 2D is a true many-body effect. How this relates to the one-particle (DFT or TB) approach?*

Reply: We thank the referee for bringing up this important point. Here we emphasize that although we used Lieb lattice mapping, the origin of FM in our model is different from the one discussed in Ref. 20, 30, 31. In those earlier studies, the FM is induced by many-body effect associated with the topological flat band (totally quenched kinetic energy) of an **ideal** Lieb lattice (*i.e.*, the middle flat band in Fig. 1b) within the Hubbard model. In contrast, the FM is a Stoner type of itinerant magnetism induced by exchange interaction for a narrowed Dirac band of a **non-ideal** Lieb lattice with large electron inhomogeneity (*i.e.*, the top localized “Dirac band” in Fig. 2a), which would be very dispersive in an ideal Lieb lattice (*i.e.*, the top Dirac band in Fig. 1b). It is for this reason, the FM we reveal here can be captured by a standard DFT calculation, similar to treating other conventional itinerant FM metals.

Comment #7: *In conclusion, I think that a standard but much more accurate analysis of DFT results will be enough to explain the origin of spin unbalance upon doping. After a deep revision, the paper should be submitted to a more appropriate journal.*

To this end I add some minor comments:

- a) Attributing the name U to the energy separation between up and down bands is misleading (U is for on-site e - e repulsion, better to use J).*
- b) The sentence “it is well known...” that appears many times in the manuscript should be accompanied with appropriate references*
- c) Figure 1a and 1b resembles too much fig 1. Of ref. 37*

Reply: As we explained in our response to the first comment, the DFT calculations alone will show spin unbalance upon doping, but cannot “explain” the origin of spin polarization.

- (a) We thank the reviewer for the suggestion. Indeed, the usage of U here is misleading, which usually denotes the on-site Coulomb energy as in the Lieb’s theorem of flat-band ferromagnetism. Different from the conventional flat-band ferromagnetism, an unconventional mechanism due to the exchange splitting of highly localized Dirac band is discovered here, where parameter J should be used rather than U . We have modified the manuscript accordingly.
- (b) We have added appropriate references as the reviewer suggested.

(c) We see the similarity between the figures in term of the colors chosen, but the content is clearly different between our Fig. 1 and the Fig. 1 of ref. 37.

Response to Reviewer #2:

Comment #1: *The authors presented theoretical investigation of the organic magnets with ferromagnetism in a two-dimensional sp^2 carbon-conjugated covalent-organic frameworks (sp^2c -COF) upon iodine doping without transition metals. By using the first-principles and tight-binding calculation, they show that the sp^2c -COF belongs to a Lieb lattice with strong electronic inhomogeneity (EI), as characterized with a large on-site energy difference and a dimerization interaction between the corner and edge-center ligands. The EI drives the usual dispersive Dirac (valence) band corresponding to the ideal Lieb lattice to become highly localized, and hence subjecting to an instability against spin polarization upon hole doping. Higher spin density for thin film and monolayer sp^2c -COF to accommodate a higher doping concentration with reduced interlayer interaction is predicted. I would suggest publication of the manuscript in Nature Communication. Some comments are provided below for the authors' consideration.*

Reply: We thank the reviewer for his/her very positive assessment and recommendation of our work.

Comment #2: *The organic ferromagnetic COF should have a finite Curie temperature, the authors may discuss its magnitude or compare it with the experimental report in the literature.*

Reply: We thank the reviewer for the suggestion. We have now added Curie temperature analysis in the revised supplementary information in comparison with experiments. The Curie temperature we obtained from DFT calculation (~9.3K) is comparable with the experimental result (~8.1K).

Comment #3: *What's the direction of the magnetic moments in the organic magnets?*

Reply: We thank the reviewer for bringing up this point. We have now added the non-collinear calculation results: “Based on non-collinear spin calculations, we found the magnetic moments prefer to align in the direction perpendicular to the plane (see details in Supplementary).”

Comment #4: *They found a higher spin density can be realized in monolayer sp^2c -COF, as the interlayer interaction is eliminated to make the valence band more localized. They also note that there is also a noticeable dispersion along the k_z direction, indicating strong interactions between the layers, which reduce the band gap to around 1.0 eV. Did the authors consider the stacking effect carefully? I guess a simple AA type stacking is used in the present work, could this simplification be justified? Same question can be asked for both cases of the bilayer and bulk structures.*

Reply: We thank the reviewer for bringing up this interesting point. The reviewer is right, we carried out calculation with a simple AA type stacking, because this is the stacking configuration that experiments observed. We agree with the reviewer that stacking may affect the system because of the strong interactions between layers, which however will not be discussed in this paper due to the reason mentioned above. Same consideration is applied for bilayer and bulk.

We added the following sentence in the method section to clarify this point: “Both sp^2c -COF and C=N-COF are reported to have AA stacking experimentally, which will be used for our study of both bulk and bilayer systems.”

Comment #5: *It’s interesting to study the enhanced magnetization upon hole doping, one may wonder what the results will be if the system is electron doped. Page 12, should B3LTP functional be B3LYP functional?*

Reply: We thank the reviewer for the point. We calculated the electron doping case, where the system remains non-magnetic. This is because the lowest conduction band is rather dispersive. We also thank the reviewer for spotting the typo of B3LTP, which should be B3LYP.

Response to Referee #3:

Comment #1: *The manuscript “Organic Ferromagnetism in a Covalent-Organic Framework” describes a purely computational study of magnetism in a covalent organic framework. The fascinating aspect about this material is that it is two-dimensional, built up by sp^2 carbon-conjugation (sp^2c), and contains NO transition metals. The manuscript itself is well-written and the information is clearly and well-presented. The authors find that this material (referred to as sp^2c -COF) can be mapped onto a Lieb lattice, which is a theoretical two-dimensional edge-centered lattice for which only artificial realizations exist. Performing density functional theory calculations, the authors can then explain the origin of*

magnetism in this material upon iodine doping. I do appreciate the layout of the manuscript, where the authors first provide the necessary background about the theoretical Lieb lattice and then show how this relates to the material in question. Only then do they show how its magnetic properties follow upon hole-doping with iodine, making it very easy to follow the arguments of the authors.

I have some comments on the novelty of this work and some technical comments, which I will describe further below. The material itself and its ferromagnetic properties are not novel--they have been described in Science 357, 673 (2017) and I assume this paper has inspired the authors to study this material. However, this Science paper does not provide any insight or even suggestions as to the mechanism that causes this unconventional magnetism, but purely reports its finding. As such, I do see three aspects that are novel: (i) The authors find that sp^2 -c-COF is a realization of a Lieb lattice. All previously found realizations for this type of lattice are somewhat artificial, using photonic, cold-atom, or surface patterning techniques. (ii) The authors can explain the origin of this unconventional magnetism in this materials. And (iii), the authors predict that a monolayer of this material (rather than the experimentally already synthesized bulk material) will exhibit even stronger ferromagnetism due to reduced inter-layer interactions. All three points are interesting in-and-of-themselves, but the authors also argue that the understanding of such magnetism in organic materials may lead to further developments and practical applications---although the language is somewhat vague and maybe more of a conjecture. Since this manuscript has several distinct novel aspects, it may appeal to a broader audience.

Reply: We thank the reviewer for reviewing our MS and his/her overall positive assessment of our work. We especially appreciate this reviewer for helping us to further clarify the novel points of our work that may appeal to a broader audience. In accordance, we have revised the manuscript to better convey these novel points.

Comment #2: *In summary, I also have a few more technical concerns: On page 5, the authors refer to the band-structure in Fig. 2b and say that “Clearly, the intrinsic $Py(BCSB)_2$ is a non-magnetic insulator with a gap around 1.5 eV”. I assume the authors have deduced the gap from a DOS, as that information could not be reliably obtained from the band-structure in Fig. 2b. Also, the magnetic state could not be deduced from the band-structure, unless a spin-resolved band structure would be plotted.*

Reply: We thank the reviewer for this comment. The reviewer is right that the band structure in Fig. 2b is not enough to deduce a reliable band gap, as it only contains several high-symmetry k -paths within $k_z=0$ plane. The band gap was actually extracted from a full Brillouion-zone calculation with much denser k -

point meshes, shown in Fig. S2. We have now revised the sentence to “Clearly, the intrinsic $\text{Py}(\text{BCSB})_2$ is a non-magnetic insulator with a band gap around 1.0 eV (see Supplementary Fig. S2), which is ...”

Yes, the magnetic state is only deduced from the spin-resolved band structure, as plotted in Fig. 4 and Fig. S6 in Supplementary.

Comment #3: *The authors seem to only have considered collinear spin arrangements. It is conceivable that non-collinear spin arrangements may play a role?*

Reply: We thank the reviewer for bringing up this point, which has also been brought up by reviewer #2. We have now added the non-collinear calculation results: “Based on non-collinear spin calculations, we found the magnetic moments prefer to align in the direction perpendicular to the plane (see details in Supplementary).”

Comment #4: *As the authors mentioned, DFT is not the correct level of theory to deduce the band gap (at least using PBE). But, why was not a post-DFT method applied, i.e. GW, to at least test the validity of the band structure and orbitals in a few cases? GW (G0W0) is implemented in VASP and the size of system might allow such calculations. Although, I would assume that the shape of the conduction bands is probably reasonably captured by the DFT calculations the authors performed.*

Reply: Thanks for the comment. We did think about doing post-DFT calculations, but the system (~102 atoms with a big vacuum layer) is too big to do GW or HSE calculation. Fortunately, as the reviewer mentioned, the expected underestimation of gap in the non-magnetic insulating state does not affect our main conclusion on ferromagnetic metal, and the shape of conduction bands, which is not important to hole doping as we needed anyway, is known to be reasonably captured by standard DFT.

Comment #5: *On page 6, the authors say that “we have qualitatively confirmed that the three valence bands [...] are resulted from the Lieb lattice”. “Confirmed” might be an overstatement here. I would use language such as “The data suggests that the valence bands are the result of a Lieb lattice”. The band structure in Fig. 2b is significantly distorted from the ideal Lieb lattice such that other interpretations are certainly possible.*

Reply: Thanks for the suggestions. We have revised statements accordingly.

Comment #6: *From a computational perspective, I am not certain why the authors switched codes for different aspects of the project. The main calculations were performed with VASP, but the calculations on the building blocks of the Lieb lattice, i.e. the Py and BCSB molecules, are performed with Gaussian. These are two very different codes (plane waves vs. Gaussian basis functions) and (PAW potentials vs. all electron). Switching codes introduces unnecessary inconsistencies and different sources or errors that may not cancel anymore. It seems perfectly reasonable that the Py and BCSB molecules would have been modeled in VASP.*

Along the same lines as the previous point---why was the exchange-correlation functional switched between calculations? For the material itself, PBE was used. On the other hand, for the Py and BCSB molecules, B3LTP was used. This introduces inconsistencies and, at best, makes the results uninterpretable.

Reply: We agree that everything can be done by VASP. The reason we used Gaussian for molecular orbitals is to save time. We note that Gaussian calculations were only performed to “qualitatively” analyze the orbital/charge distributions, no quantitative information is extracted. Therefore, it should yield reasonable results without causing inconsistencies, including the different exchange-correlation functional, because all the DFT based methods are expected to at least agree with each other qualitatively in charge density distributions. Nevertheless, we agree with the reviewer that it is better to use the same method consistently, so we have redone those molecular orbital calculations using VASP with PAW-PBE replacing the Gaussian results. These results are indeed consistent with our previous Gaussian calculation results.

We added the following sentence in the main text: “*We have also calculated molecular information using Gaussian method showing consistent results (see Supplementary Fig. S5)*”

Comment #7: *On page 10, the authors refer to Fig. S8, but I assume they meant Fig. S7.*

Reply: It has been corrected, thanks!

Comment #8: *Why was the unit cell for sp^2c -COF not relaxed while the one for C=N-COF was?*

Reply: Originally, we used experimental lattice parameters [Science 357, 673 (2017)] for sp^2c -COF but relaxed C=N-COF, because no experimental data were provided. We have now relaxed sp^2c -COF too,

which yields a negligible change of lattice parameters (<0.1%) compared to the experimental results, and this change has negligible influence to the electronic and magnetic properties of the COF.

We added the following sentence in the methods to clarify this point: “*For the sp^2c -COF, we adopted the experimental lattice constant [full structural relaxation would only change the lattice constants negligibly (less than 0.1%)]. While for the C=N-COF without available experimental lattice constants, the structure was fully relaxed*”

Comment #9: *I assume that the technical points can be overcome by the authors so that the deciding factor will be one of novelty. As pointed out above, the main point of magnetism in sp^2c -COF is not novel, but other points made by the authors are.*

Reply: We thank the reviewer for this constructive comment to better clarify the novel points of our work. We agree that the phenomenon of the ferromagnetism of COF as already observed is not novel, while the underlying physical mechanism we discovered is novel. Especially, as the reviewer pointed out, the discovery of the first material realization of the long-sought Lieb lattice and the proposal of magnetic enhancement are novel and more interesting. Accordingly, we have substantially modified the title/abstract/introduction/conclusion to better present these novel points of our work.

“*Realization of Lieb Lattice in a Covalent-Organic Framework and Its Unconventional Ferromagnetism*” will be the new title.

Reviewers' comments:

Reviewer #1 (Remarks to the Author):

In the revised version of the manuscript, the authors provide more details on the DFT calculation. Now the description of DFT results is more accurate and meets the standards for these kind of studies.

Still, the declared goal of the paper – the identification of the sp²c-COF as the first material realization of a Lieb-lattice and the consequent explanation of the physical origin of magnetism in this system – is not achieved.

I stress again that the Tight Binding parametrization that the authors present as evidence of the Lieb-lattice behavior is intrinsically incorrect: Tight Binding parametrization is nothing else than a change of basis for the eigenstates. As such, it must reproduce all the eigenstates of the system in the energy region of interest. The orbital character of valence and conduction bands may be different but they are intrinsic and should be obtained by the same set of Tight Binding parameters. This is not the case here and therefore the identification of sp²c-COF as a Lieb-lattice is not demonstrated.

The identification of sp²c-COF as a Lieb-lattice- if demonstrated - would not add any physical insight into the origin of "unconventional" ferromagnetism in this system. The authors recall now in the introduction that the peculiar properties of a Lieb lattice is " to host exotic flat bands subject to strong correlation effects." Indeed e-e correlation is at the core of the " various intriguing quantum states, e.g., ferromagnetism 1, 2, superconductivity 3" that make a standard 2D square lattice a Lieb-lattice. Many body effects are absent both in DFT and in the Tight Binding parametrization, therefore the Lieb mechanism is out of order in the present case.

The authors answer to my criticism about their concept of "local electronic chemical potential varying from site to site" with wrong arguments: by very definition, chemical potential (synonymous of Fermi level in electronic systems) cannot change from point to point in a system at EQUILIBRIUM. Putting together two systems characterized by different chemical potentials would indeed induce a rearrangement of charges just to re-align the chemical potentials. I believe the authors do not mean that this is the case here, or their system would be out-of-equilibrium. The difference in on-site energies responsible of what they call "electron inhomogeneity" has nothing to do with the chemical potential. I find at least very surprising that the authors, instead of simply correcting this misuse of a very basic concept, decided to maintain it and with wrong arguments.

Overall, the paper is not convincing in reaching its declared goal and for this reason I think it should not be published in Nature Communications.

Reviewer #2 (Remarks to the Author):

In the present work of "Realization of Lieb Lattice in a Covalent-Organic Framework and Its Unconventional Ferromagnetism" by W. Jiang, Huaqing Huang, and Feng Liu, the authors predicted the first material realization of the Lieb lattice and investigated the ferromagnetism in the 2D organic system sp²c-COF upon iodine doping by using the DFT and TB calculations. The unconventional ferromagnetism mechanism in sp²c-COF upon iodine doping has been well understood by the strong electronic inhomogeneity. On the other hand, this unconventional mechanism for ferromagnetism offers a new theoretical guidance in the prediction of new COF-based organic magnets. In my opinion, this is a well-written manuscript and this work will motivate both experimental and theoretical work in the future. So I recommend publication in

Nature communication.

Reviewer #3 (Remarks to the Author):

The authors have addressed all my raised concerns properly and I think the manuscript is now suitable for publication in Nature Communications. It was refreshing to see that the authors did not try to argue about the various points, but simply performed the additional calculations to prove/disprove concerns raised.

Response to reviewer #1's comments:

Comment #1: *I stress again that the Tight Binding parametrization that the authors present as evidence of the Lieb-lattice behavior is intrinsically incorrect: Tight Binding parametrization is nothing else than a change of basis for the eigenstates. As such, it must reproduce all the eigenstates of the system in the energy region of interest. The orbital character of valence and conduction bands may be different but they are intrinsic and should be obtained by the same set of Tight Binding parameters. This is not the case here and therefore the identification of sp^2c -COF as a Lieb-lattice is not demonstrated.*

Reply: We are sorry that there seemed some confusion by this reviewer. Our TB model and parametrization do produce all the eigenstates in the energy range of interest, i.e., the eight bands around Fermi level (five above and three below) without hole doing. They are indeed obtained by the same set of TB parameters, including eight molecular orbitals on eight lattice sites, which can be formally expressed in an eight-band TB Hamiltonian, as shown below and now also added in the SM. It should be noted that these eight bands happened to be decomposed into two sets of bands, because of the negligible interaction between them, as evidenced from DFT calculations. Specifically, DFT wavefunction analysis shows that these two sets of bands come from orbital basis of two sub lattices with little orbital overlap and large energy separation. Consequently, the 8-band TB Hamiltonian can be block diagonalized into two blocks of 3- and 5-bands each (marked as red blocks in the figure below). It is for this reason that the 8-band TB model is effectively reduced to one 3-band and one 5-band model, which are further revealed to correspond to a Lieb-3 and a Lieb-5 band, respectively. Note also that our discussion has focused only on the Lieb-3 band because upon hole doing the Fermi level is moved into the top band of these three band to induce the magnetization of our interest. We hope that our explanation will mitigate the reviewer's concerns on this point. Our TB model and parametrization are correct, which indeed enabled us to identify sp^2c -COF as a Lieb-lattice material.

$$H = \begin{pmatrix} \boxed{\begin{matrix} E_0 + \Delta E_0 & t_0 \cdot V_{c-ex} & t_0 \cdot V_{c-ey} \\ t_0 \cdot V_{c-ex}^* & E_0 & t'_0 \cdot V_{ex-ey} \\ t_0 \cdot V_{c-ey}^* & t'_0 \cdot V_{ex-ey}^* & E_0 \end{matrix}} & \boxed{\begin{matrix} 0 & 0 & 0 & 0 & 0 \\ 0 & 0 & 0 & 0 & 0 \\ 0 & 0 & 0 & 0 & 0 \end{matrix}} \\ \boxed{\begin{matrix} 0 & 0 & 0 \\ 0 & 0 & 0 \\ 0 & 0 & 0 \\ 0 & 0 & 0 \\ 0 & 0 & 0 \end{matrix}} & \boxed{\begin{matrix} E_1 + \Delta E_1 & t_1 \cdot V_{c-ex} & t_1 \cdot V_{c-ex}^* & t_1 \cdot V_{c-ey} & t_1 \cdot V_{c-ey}^* \\ t_1 \cdot V_{c-ex}^* & E_1 & t_1 \cdot V_{ex-ex} & t'_1 \cdot V_{ex-ey} & t'_1 \cdot V_{ey-ex} \\ t_1 \cdot V_{c-ex} & t_1 \cdot V_{ex-ex}^* & E_1 & t'_1 \cdot V_{ey-ex}^* & t'_1 \cdot V_{ex-ey}^* \\ t_1 \cdot V_{c-ey}^* & t'_1 \cdot V_{ex-ey}^* & t'_1 \cdot V_{ey-ex} & E_1 & t_1 \cdot V_{ey-ey} \\ t_1 \cdot V_{c-ey} & t'_1 \cdot V_{ey-ex}^* & t'_1 \cdot V_{ex-ey} & t_1 \cdot V_{ey-ey}^* & E_1 \end{matrix}} \end{pmatrix}$$

Comment #2: *The identification of sp^2c -COF as a Lieb-lattice- if demonstrated - would not add any physical insight into the origin of “unconventional” ferromagnetism in this system. The authors recall now in the introduction that the peculiar properties of a Lieb lattice is “to host exotic flat bands subject to strong correlation effects.” Indeed e - e correlation is at the core of the “ various intriguing quantum states, e.g., ferromagnetism^{1,2}, superconductivity³“ that make a standard 2D square lattice a Lieb-lattice. Many body effects are absent both in DFT and in the Tight Binding parametrization, therefore the Lieb mechanism is out of order in the present case.*

Reply: This point raised by the reviewer is somewhat semantic, namely, what should be called the Lieb mechanism or Lieb physics. In our opinion it is not critical, certainly not as important as the new physics we have shown here. One may consider any band structure derived from a Lieb lattice (i.e., an edge-centered square lattice), either ideal or not ideal, can be generally called Lieb band and the physics (mechanism) associated with such band can be called Lieb physics (mechanism). While the reviewer suggests only the many-body physics associated with the flat band of an ideal Lieb lattice at half filling should be called Lieb mechanism. In fact, we do not necessarily object this notion. We would like to point out to the reviewer that we never called the ferromagnetism we discover as the “Lieb mechanism”. Instead we clearly stated it differently as an unconventional ferromagnetism in a non-ideal “Lieb lattice”. Our work is to identify sp^2c -COF as a Lieb-lattice material exhibiting unconventional ferromagnetism due to electronic inhomogeneity, which differs from the original Lieb mechanism of ferromagnetism due to many-body correlation effect. In this regard, our work **significantly enriches** the physics of Lieb lattice, which we believe is very interesting. Apparently, our view is shared by both Reviewer #2 and #3 who recommended publication of our paper in Nature Communications.

Comment #3: *The authors answer to my criticism about their concept of “local electronic chemical potential varying from site to site” with wrong arguments: by very definition, chemical potential (synonymous of Fermi level in electronic systems) cannot change from point to point in a system at EQUILIBRIUM. Putting together two systems characterized by different chemical potentials would indeed induce a rearrangement of charges just to re-align the chemical potentials. I believe the authors do not mean that this is the case here, or their system would be out-of-equilibrium. The difference in on-site energies responsible of what they call “electron inhomogeneity” has nothing to do with the chemical potential. I find at least very surprising that the authors, instead of simply correcting this misuse of a very basic concept, decided to maintain it and with wrong arguments.*

Reply: Actually we did mean that the final electronic (band) structure at equilibrium with one Fermi level (chemical potential) can be thought as formed by bringing together two sub-lattice systems with

different chemical potentials (i.e., on-site or molecular level energies) interact with each other through electronic hopping and charge transfer. It is in analogy to formation of a p-n junction with an equal chemical potential at equilibrium, which is formed by bringing together two sub-systems of p- and n-type with different chemical potentials (i.e. defect level energies) interacting with each other. Since this may cause confusion as indicated by the reviewer, we now opt to simply remove it, replacing “local chemical potential difference” with “on-site energy difference”, without possible complication.

Reviewers' comments:

Additional Reviewer#1 (Remarks to the Author):

In this work entitled 'Realization of Lieb Lattice in a Covalent-Organic Framework and Its Unconventional Ferromagnetism', the authors provide interesting insights into the origin of ferromagnetism in the special sp²c-COF via theoretical investigation, unrevealing the hidden physical picture which are not fully discussed in the experimental works such as Science 357, 673 (2017). From this point of view, I find this work make good sense to a broad range of researchers.

After carefully reading, in my opinion, the criticism by Review #1 are mainly focused on the terminology and may not change the physics discussed by this work. As stated by Review #1, the parameters in tight-binding methods are usually empirical and fitted by experimental or first-principle calculations. But this does mean that tight-binding cannot capture the true physics. I have one major concern and one minor concern here. Minor revision may be required to make their work more clear, and also to partially address Reviewer #1's confusion.

1. The major concern is:

Interestingly, just like graphene could be viewed as the experimental realization of an artificial hexagonal lattice, the authors claim here sp²c-COF is a realization of artificial Lieb lattice. And what's more, sp²c-COF shows an unconventional ferromagnetism compared with the standard artificial Lieb lattice. For clarity,

A Lieb lattice: Dirac band + flat band (where ferromagnetism comes from, edge states)

sp²c-COF: Dirac band (first VB, when doping, EI-induced ferromagnetism) + flat band (second VB, edge states)

Here arouses the confusion. As reported by the authors in the main context from P2, L28-L33, 'a two-dimensional (2D) edge-centered square lattice (Fig. 1a) is one of a few lattices hosting exotic flat bands subject to strong correlation effects. It has been proposed to spawn various intriguing quantum states, e.g., ferromagnetism, superconductivity, and topological states. More recently, it has been further predicted to hold intriguing optical, electronic, and magnetic states.' We find that if the flat band in Lieb lattice is one of the reason to account for its various intriguing quantum states. In sp²c-COF, the flat band is even not the one to account for its ferromagnetism, how should we expect more intriguing quantum states in such 'Lieb lattice' like sp²c-COF? I think the authors should concern more on this point, or we can't be convinced with the claimed importance of this 'unconventional ferromagnetism' by the authors.

2. The minor concern is:

By saying 'realization of Lieb lattice...' in the title is somewhat misleading, since the realization of sp²c-COF was reported in Science 357, 673 (2017), and if demonstrated, such sp²c-COF is at least topologically analogical to the Lieb Lattice.

This work is well written with a good logic, and I appreciate the authors' good instinct to recognize sp²c-COF as a Lieb lattice then unraveled the hidden origin of ferromagnetism observed experimentally in sp²c-COF, although it still needs more evidence to equal sp²c-COF to a Lieb lattice, when concerning the as-claimed intriguing quantum states in such systems. If the above concerns are well-addressed, I will recommend publication of this work in nature communications.

Additional Reviewer#2 (Remarks to the Author):

I have been asked to express my opinion on the paper from Jiang and co-workers after two rounds of review. The manuscript, titled "Realisation of Lieb lattice in a covalent-organic framework and its unconventional ferromagnetism" has two main goals: 1) to demonstrate that a recent synthesised covalent-organic framework (COF) represents a physical realisation of the Lieb lattice, and 2) to show that hole doping can induce a magnetic ground state in such lattice. The authors tackle this problem by a combination of DFT and tight-binding model. In particular they map the DFT calculations onto a tight-binding Hamiltonian in order to show that the electronic structure is that of a Lieb lattice. Then they use DFT alone to show the magnetism and the TB to interpret it.

In general I believe that the evidence brought forward is enough to make the case. I am not 100% convinced about the broad appeal of the manuscript, which appears to me as a nice curiosity but I cannot see too much potential for any significant further development. However, since the other referees do not question the broad appeal of the manuscript I would restrain myself to do so. As such I think that the manuscript may be publishable. However, before doing so, I'd like the authors to address a few points:

1) I think that there is a bit of confusion in the explanation of the mechanism for the ferromagnetism, a confusion that also emerged from the discussion with one of the referees. In particular this concerns the role of interaction in relation to the formation of flat bands. For what I can see the origin seems to be clear: magnetism arises from the Stoner criterion acting on a flat band made of 2p states. Since the Stoner parameter for the p shell is large, in fact larger than that of the 3d shell [see J.F. Janak, Phys. Rev. B 16, 255 (1977)], the band spin splits. What the authors then call "a novelty" is that the band is flat not because of a strong Coulombic repulsion (or a small t), but because of the Lieb topology of the lattice. This seems to me simple enough to explain and can be remarked early on in the manuscript.

2) There is a simple way to extract the tight-binding Hamiltonian from the DFT calculations, namely that of performing a maximally localised Wannier functions transformation [Computer Physics Communications 178 (9), 685-699 (2008)]. Such method is available for VASP. I believe that the paper would be on a much more solid ground if such calculation is performed.

3) I am not sure I understand how the authors attribute the various bands. They claim that the valence band is mainly derived from p_x and p_y orbitals, but then they discuss this as a pi-conjugated band. For what I can see it is a sigma band instead (the p orbitals are in plane), while the conduction one is pi-conjugated. Can the authors clarify this point?

4) I am puzzled by the claim that non-collinear calculations reveal that the magnetisation is perpendicular to the plane. This cannot be correct. If I understand correctly the calculations do not include spin-orbit coupling. If this is the case the system is SU(2) invariant and therefore there is no preferential orientation for the magnetisation (all directions must be energy degenerate).

In conclusion I think that the paper provides enough evidence for explaining the ferromagnetism in terms of Stoner criterion on Lieb-lattice-like bands. As such it can be published. However, the authors have first to respond to my comments.

We thank both reviewers for reviewing our manuscript and providing an overall positive assessment of our work. We also thank them for providing some helpful and constructive comments, which have enabled us to further improve the clarity and quality of the paper. Below we respond in detail to their comments in a point-by-point manner.

Response to reviewer #1's comments:

Comment #1: *In this work entitled 'Realization of Lieb Lattice in a Covalent-Organic Framework and Its Unconventional Ferromagnetism', the authors provide interesting insights into the origin of ferromagnetism in the special sp²c-COF via theoretical investigation, unrevealing the hidden physical picture which are not fully discussed in the experimental works such as Science 357, 673 (2017). From this point of view, I find this work make good sense to a broad range of researchers.*

After carefully reading, in my opinion, the criticism by Reviewer #1 are mainly focused on the terminology and may not change the physics discussed by this work. As stated by Reviewer #1, the parameters in tight-binding methods are usually empirical and fitted by experimental or first-principle calculations. But this does mean that tight-binding cannot capture the true physics. I have one major concern and one minor concern here. Minor revision may be required to make their work more clear, and also to partially address Reviewer #1's confusion.

Reply: We thank the reviewer for a fair, insightful and constructive review.

Comment #2: *1. The major concern is: Interestingly, just like graphene could be viewed as the experimental realization of an artificial hexagonal lattice, the authors claim here sp²c-COF is a realization of artificial Lieb lattice. And what's more, sp²c-COF shows an unconventional ferromagnetism compared with the standard artificial Lieb lattice. For clarity, A Lieb lattice: Dirac band + flat band (where ferromagnetism comes from, edge states); sp²c-COF: Dirac band (first VB, when doping, EI-induced ferromagnetism) + flat band (second VB, edge states). Here arouses the confusion. As reported by the authors in the main context from P2, L28-L33, 'a two-dimensional (2D) edge-centered square lattice (Fig. 1a) is one of a few lattices hosting exotic flat bands subject to strong correlation effects. It has been proposed to spawn various intriguing quantum states, e.g., ferromagnetism, superconductivity, and topological states. More recently, it has been further predicted to hold intriguing optical, electronic, and magnetic states.' We find that if the flat band in Lieb lattice is one of the reason to account for its various intriguing quantum states. In sp²c-COF, the flat band is even not the one to*

account for its ferromagnetism, how should we expect more intriguing quantum states in such 'Lieb lattice' like sp²c-COF? I think the authors should concern more on this point, or we can't be convinced with the claimed importance of this 'unconventional ferromagnetism' by the authors.

Reply: Thanks for this point of clarification. We have now revised the related statements to clearly demonstrate the unconventional ferromagnetism in our non-ideal Lieb-like lattice arising from charge inhomogeneity is different from the conventional ferromagnetism in an ideal Lieb lattice arising from instability of flat bands, as following:

1. Change the title to “*A Lieb-Like Lattice in ...*” (see also reply to the reviewer #2's comment #2 below)
2. In the introduction (Page 3 Paragraph 3): “*The magnetism we show here arises from a localized state converted by EI from an otherwise dispersive state in the ideal Lieb lattice. Consequently, we call it unconventional because conventional magnetism usually arises from the default localized states with a strong Columbic repulsion (or a small hopping t), such as the flat bands in ideal 2D Lieb and Kagome lattices, and the localized orbitals in organic compounds and nano-carbon structures.*”

Comment #3: *2. The minor concern is: By saying 'realization of Lieb lattice...' in the title is somewhat misleading, since the realization of sp²c-COF was reported in Science 357, 673 (2017), and if demonstrated, such sp²c-COF is at least topologically analogical to the Lieb Lattice.*

Reply: Thanks. We have now changed the title from “*Realization of Lieb Lattice in a Covalent-Organic Framework ...*” to “*A Lieb-Like Lattice in a Covalent-Organic Framework ...*”.

Comment #4: *This work is well written with a good logic, and I appreciate the authors' good instinct to recognize sp²c-COF as a Lieb lattice then unraveled the hidden origin of ferromagnetism observed experimentally in sp²c-COF, although it still needs more evidence to equal sp²c-COF to a Lieb lattice, when concerning the as-claimed intriguing quantum states in such systems. If the above concerns are well-addressed, I will recommend publication of this work in nature communications.*

Reply: Again, we thank the reviewer for a fair, insightful and constructive review.

Response to reviewer #2's comments:

Comment #1: *I have been asked to express my opinion on the paper from Jiang and co-workers after two rounds of review. The manuscript, titled “Realization of Lieb lattice in a covalent-organic framework and its unconventional ferromagnetism” has two main goals: 1) to demonstrate that a recent synthesized covalent-organic framework (COF) represents a physical realization of the Lieb lattice, and 2) to show that hole doping can induce a magnetic ground state in such lattice. The authors tackle this problem by a combination of DFT and tight-binding model. In particular they map the DFT calculations onto a tight-binding Hamiltonian in order to show that the electronic structure is that of a Lieb lattice. Then they use DFT alone to show the magnetism and the TB to interpret it.*

In general I believe that the evidence brought forward is enough to make the case. I am not 100% convinced about the broad appeal of the manuscript, which appears to me as a nice curiosity but I cannot see too much potential for any significant further development. However, since the other referees do not question the broad appeal of the manuscript I would restrain myself to do so. As such I think that the manuscript may be publishable. However, before doing so, I'd like the authors to address a few points:

Reply: We thank the reviewer for a fair, insightful and constructive review.

Comment #2: *1) I think that there is a bit of confusion in the explanation of the mechanism for the ferromagnetism, a confusion that also emerged from the discussion with one of the referees. In particular this concerns the role of interaction in relation to the formation of flat bands. For what I can see the origin seems to be clear: magnetism arises from the Stoner criterion acting on a flat band made of 2p states. Since the Stoner parameter for the p shell is large, in fact larger than that of the 3d shell [see J.F. Janak, Phys. Rev. B 16, 255 (1977)], the band spin splits. What the authors then call “a novelty” is that the band is flat not because of a strong Coulombic repulsion (or a small t), but because of the Lieb topology of the lattice. This seems to me simple enough to explain and can be remarked early on in the manuscript.*

Reply: We thank the reviewer for pointing out this important point. Indeed, the ferromagnetism arises from the Stoner criterion acting on a flat band made of 2p states. We have now revised the manuscript and remark the novelty early on in the manuscript with the proper references added:

Page3, Paragraph3: **“It is important to note that this unique EI-induced magnetism in an organic lattice is different from those exotic mechanisms proposed before. The magnetism we show here arises from a**

localized state converted by EI from an otherwise dispersive state in the ideal Lieb lattice. Consequently, we call it unconventional because conventional magnetism usually arises from the default localized states with a strong Columbic repulsion (or a small hopping t), such as the flat bands in 2D Lieb and Kagome lattices, and the localized orbitals in organic compounds and nano-carbon structures.”

Page7, Paragraph2: “Because the VB is made of $2p$ state, the Stoner parameter is in fact even larger than that of the $3d$ state [J.F. Janak, *Phys. Rev. B* 16, 255 (1977)]. Consequently, one may expect that this band will now be subjected to an instability against spin-polarization upon hole doping.”

Comment #3: *2) There is a simple way to extract the tight-binding Hamiltonian from the DFT calculations, namely that of performing a maximally localised Wannier functions transformation [Computer Physics Communications 178 (9), 685-699 (2008)]. Such method is available for VASP. I believe that the paper would be on a much more solid ground if such calculation is performed.*

Reply: Thanks for the suggestion. We have further performed maximally localized Wannier functions fitting using Wannier90 package, which show very good agreement with our previous DFT calculation results and tight-binding analysis. These results are added into the main text to support our theory with the proper references cited, as follows:

Page6, Paragraph2: “To more concretely confirm the Lieb-lattice nature of the Py(BCSB)2, we performed the maximally localized Wannier functions fitting using the Wannier90 package [Computer Physics Communications 178 (9), 685-699 (2008)]. The fitted band structure and the corresponding maximally localized Wannier functions show good consistency with the above DFT calculation results and TB analyses below (see Supplementary Fig. S6).”

Page 7, Paragraph1: “We then compared our TB fitting parameters with those of the Wannier fitted Hamiltonian, which shows a very good agreement, further confirming the strong EI effect to induce a localization of the VB (see Supplementary).”

Comment #4: *3) I am not sure I understand how the authors attribute the various bands. They claim that the valence band is mainly derived from p_x and p_y orbitals, but then they discuss this as a pi-conjugated band. For what I can see it is a sigma band instead (the p orbitals are in plane), while the conduction one is pi-conjugated. Can the authors clarify this point?*

Reply: Sorry for the confusion. In Fig.2, we showed the band contribution from both px/py and pz orbitals (labeled by orange and black circles, respectively), where the valence and conduction bands are all mainly contributed by pz orbitals (big black circles), *i.e.*, pi-conjugated bands. We realized that such confusion may arise from our different coloring for the valence (red) and conduction (blue) bands, where the valence band color may be confused with the orange labeling for px/py orbitals. To avoid further confusion, we now changed the orange color into green color.

Comment #5: *4) I am puzzled by the claim that non-collinear calculations reveal that the magnetisation is perpendicular to the plane. This cannot be correct. If I understand correctly the calculations do not include spin-orbit coupling. If this is the case the system is SU(2) invariant and therefore there is no preferential orientation for the magnetisation (all directions must be energy degenerate).*

Reply: We thank the reviewer for bringing up this point. In our calculation, we included the spin-orbit coupling (SOC) and use the non-collinear calculation to initiate different magnetization directions. We realize that such confusion is coming from our unclear method description, where only non-collinear calculation is mentioned without SOC. Now we have modified method description in the main text as well as in the supplementary, as follows:

Page8, Paragraph2: “Using non-collinear spin calculation considering spin-orbit coupling, we found ...”

Comment #6: *In conclusion I think that the paper provides enough evidence for explaining the ferromagnetism in terms of Stoner criterion on Lieb-lattice-like bands. As such it can be published. However, the authors have first to respond to my comments.*

Reply: Again, we thank the reviewer for a fair, insightful and constructive review. We have now made all the revisions as suggested, to further clarify our work.

REVIEWERS' COMMENTS:

Reviewer #4 (Remarks to the Author):

This is my second review of the manuscript of Jiang and co-workers. In their reply the authors have addressed all my criticisms in a thorough and convincing way. As such the manuscript has significantly improved and it is now suitable for publication.